# Psychometric Properties of the German Version of the Rivermead Post-Concussion Symptoms Questionnaire in Adolescents after Traumatic Brain Injury and Their Proxies

**DOI:** 10.3390/jcm12010319

**Published:** 2022-12-31

**Authors:** Fabian Bockhop, Marina Zeldovich, Sven Greving, Ugne Krenz, Katrin Cunitz, Dagmar Timmermann, Elena M. Bonke, Michaela V. Bonfert, Inga K. Koerte, Matthias Kieslich, Maike Roediger, Michael Staebler, Steffen Berweck, Thomas Paul, Knut Brockmann, Philine Rojczyk, Anna Buchheim, Nicole von Steinbuechel

**Affiliations:** 1Institute of Medical Psychology and Medical Sociology, University Medical Center Göttingen, 37073 Göttingen, Germany; 2Department of Child and Adolescent Psychiatry, Psychosomatics, and Psychotherapy, Ludwig-Maximilians-Universität München, 80336 Munich, Germany; 3Department of Paediatric Neurology and Developmental Medicine, Dr. von Hauner Children’s Hospital, Ludwig-Maximilians-Universität München, 80337 Munich, Germany; 4Department of Paediatric Neurology, Hospital of Goethe University, 60590 Frankfurt am Main, Germany; 5Department of Pediatrics and Adolescent Medicine, General Pediatrics, Division of Neuropediatrics, University Hospital Muenster, 48149 Muenster, Germany; 6Neurological Rehabilitation Center for Children, Adolescents and Young Adults, 78262 Gailingen am Hochrhein, Germany; 7Specialist Center for Paediatric Neurology, Neurorehabilitation and Epileptology, Schoen Klinik Vogtareuth, 83569 Vogtareuth, Germany; 8Department of Pediatric Cardiology, Neonatology, Intensive Care Medicine and Pneumology, University Medical Center Göttingen, 37075 Göttingen, Germany; 9Interdisciplinary Pediatric Center for Children with Developmental Disabilities and Severe Chronic Disorders, Department of Pediatrics and Adolescent Medicine, University Medical Center Göttingen, 37075 Göttingen, Germany; 10Institute of Psychology, Faculty of Psychology and Sport Science, University of Innsbruck, 6020 Innsbruck, Austria

**Keywords:** pediatric traumatic brain injury, Rivermead Post-Concussion Symptoms Questionnaire, psychometric properties, classical test theory

## Abstract

The Rivermead Post-Concussion Symptoms Questionnaire (RPQ) assesses post-concussion symptoms (PCS) after traumatic brain injury (TBI). The current study examines the applicability of self-report and proxy versions of the German RPQ in adolescents (13–17 years) after TBI. We investigated reliability and validity on the total and scale score level. Construct validity was investigated by correlations with the Post-Concussion Symptoms Inventory (PCSI-SR13), Generalized Anxiety Disorder Scale 7 (GAD-7), and Patient Health Questionnaire 9 (PHQ-9) and by hypothesis testing regarding individuals’ characteristics. Intraclass correlation coefficients (ICC) assessed adolescent–proxy agreement. In total, 148 adolescents after TBI and 147 proxies completed the RPQ. Cronbach’s α (0.81–0.91) and McDonald’s ω (0.84–0.95) indicated good internal consistency. The three-factor structure outperformed the unidimensional model. The RPQ was strongly correlated with the PCSI-SR13 (self-report: r = 0.80; proxy: r = 0.75) and moderately–strongly with GAD-7 and PHQ-9 (self-report: r = 0.36, r = 0.35; proxy: r = 0.53, r = 0.62). Adolescent–proxy agreement was fair (ICC [2,1] = 0.44, CI_95%_ [0.41, 0.47]). Overall, both self-report and proxy assessment forms of the German RPQ are suitable for application in adolescents after TBI. As proxy ratings tend to underestimate PCS, self-reports are preferable for evaluations. Only if a patient is unable to answer, a proxy should be used as a surrogate.

## 1. Introduction

Traumatic brain injury (TBI) has been identified as the primary injury-related cause of death and disability in children and adolescents worldwide [2]. Reports of the global incidence of pediatric TBI range from 47 to 280 per 100.000 children/adolescents but vary greatly across countries and injury causes [1], with road traffic accidents, incidental falls, and other accidents being the most common causes of injury in Europe [3]. Pediatric TBI is associated with profound short- and long-term functional [4], cognitive [5], as well as developmental and behavioral [6] impairments. Consequently, pediatric TBI poses a substantial burden on injury victims, caregivers, as well as global societies at large [7].

Post-concussion symptoms (PCS) are one of the most frequent sequelae after pediatric TBI. They include headaches, pain in various areas of the body, nausea, light- and noise-sensitivity, double vision, trouble focusing, as well as fatigue among others [8]. The prevalence of PCS in pediatric population ranges widely (from 24.5% to 52%) depending on the diagnostic criteria and the time since injury [9]. Most acute PCS are resolved within a few months after injury [10], however chronic symptoms may be observed over a number of years post TBI [11]. Children and adolescents with persistent PCS report lower health-related quality of life (HRQOL) compared with those who recovered fully after injury, whereby such deficits remain observable after PCS have resolved [12].

PCS screenings are routinely based on self-report instruments that do not require advanced technical equipment or in-depth practice for their administration [13]. One of the most commonly used patient-reported outcome measures (PROM) is the Rivermead Post-Concussion Symptoms Questionnaire (RPQ) [14]. The RPQ covers a total of 16 items (headaches, dizziness, nausea and/or vomiting, noise sensitivity, sleep disturbance, fatigue, irritability, depression, frustration, forgetfulness and poor memory, poor concentration, slow thinking, blurred vision, light sensitivity, double vision, and restlessness). The presence and the severity of these PCS is rated relative to the individuals’ subjective condition before TBI based on a five-point Likert scale (0 = ‘not present at all’, 1 = ‘no more of a problem than before’, 2 = ‘a mild problem’, 3 = ‘a moderate problem’, to 4 = ‘severe problem’). The original English version designed for adults after TBI demonstrated good psychometric properties and has been proposed to be reliable both as a self-report or a clinician-administered measure [14]. In the Anglo-Saxon context, previous studies have predominantly focused on young, concussed athletes (14–20 years) and provided only limited information on the reliability and the validity of the instrument [15,16,17,18]. The authors have reported good reliability (Cronbach’s α = 0.89; split-half reliability r = 0.88), low test–retest reliability (r_tt_ = 0.23 to r_tt_ = 0.24) [15], and good discriminative ability between concussed individuals and controls (Cohen’s d = 1.57) [16]. However, despite its repeated application after pediatric TBI [17,18], the RPQ is listed in the Common Data Elements (CDE) recommendations as a supplemental measure only for use in adults [19].

For PCS assessment in children and adolescents, the CDE recommendations [19] propose the use of the Post-Concussion Symptom Inventory (PCSI) [20,21]. The PCSI is available in age-adjusted pre-TBI and post-TBI forms for children (aged 8–12 years; PCSI-SR8) and adolescents (aged 13–17 years; PCSI-SR13) as well as in a proxy version (e.g., filled in by parents or caregivers) [21]. The understandability of the PCSI forms was investigated in the respective age groups using cognitive debriefing [21]. The PCSI-SR13 is comparable to the RPQ in terms of item content and wording, with the main difference being the form of administration. While the PCSI-SR13 self-report and proxy versions have two separate forms assessing the experience of PCS before and after TBI, the response scale rating of the RPQ comprises both pieces of information jointly.

The evaluation of the psychometric properties of the RPQ in adolescents and their proxies is crucial for its application in lifelong clinical follow-up and multicenter pediatric studies for several reasons. First, the repeated application of the RPQ in pediatric studies has resulted in reports of reasonable prevalence rates of acute PCS following mild TBI based on self-reports [18] as well as combined proxy versions [22]. Hence, the validation of the RPQ would further strengthen the reliability of these previous results for conclusions in the clinical context. Second, good to excellent psychometric properties of the instrument have been demonstrated across several translations in adults after TBI – including the German version [23,24]. As these findings provide solid evidence in favor of the use of the RPQ in multinational research and clinical practice, further validation would suggest a similarly widespread applicability of this instrument in adolescents. Third, the administration of a single RPQ form covering both pre- and post-TBI symptoms allows for the time-efficient and low-cost assessment of PCS, thereby enabling and simplifying longitudinal PCS assessment across the lifespan. The RPQ for adolescents would then be applicable during an important period in an individual’s life. Finally, the investigation of the validity of proxy assessments can provide deeper insights into the usability of the RPQ as an observer rating of PCS in adolescents after TBI. Proxy assessments could be used as a surrogate for self-reports, whenever the adolescents themselves are not able to report.

To date, there have not been any systematic investigations of the psychometric properties of either the original English version or other translations of the RPQ in pediatric samples. The aim of the present study is, therefore, to investigate the classical psychometric properties of the self-report and proxy versions of the German RPQ for the assessment of PCS in adolescents after TBI aged 13–17 years. The respective analyses include examinations of reliability, factorial validity, and construct validity. Evidence of acceptable psychometric properties would suggest the reliable and valid assessment of the PCS in adolescents after TBI.

## 2. Materials and Methods

### 2.1. Study Sample

The present study was conducted as part of the Quality of Life after Brain Injury for Kids and Adolescents (QOLIBRI-Kid/Ado) project, which aimed to develop the first disease-specific, age-adapted HRQOL instrument for children (8–12) and adolescents (13–17 years) after TBI. The study sample was a retrospective, clinical convenience sample. Participants were recruited from twelve medical centers and hospitals in Germany between January 2019 and January 2022 based on a procedure consisting of multiple steps. Recruitment sites provided patient lists which included pediatric patients with a diagnosis of TBI (ICD code S06*) of the past ten years. Potential participants were selected and invited by mail for study participation based on inclusion and exclusion criteria. After they had given their informed consent, appointments were arranged for online or in-person assessments in the centers. The inclusion criteria were an age of 8–17 years, the date of TBI diagnosis being at least three months but not more than ten years before study enrollment, available information on the TBI severity (either based on the Glasgow Coma Scale [GCS] [25] or medical reports), as well as the ability to understand and fill in questionnaires. Children and adolescents who suffered from severe epilepsy, severe polytrauma, serious mental illness prior to TBI, or disease leading to death were excluded from this study.

In total, more than 5000 families were contacted. Based on sample size estimates within the QOLIBRI-Kid/Ado study, at least 140 participants and their proxies had to be included for each age group [26]. In the end, *n* = 300 subjects (8–12 years: *n* = 152; 13–17 years: *n* = 148) were recruited as participants. Children and adolescents were tested online (8–12 years: *n* = 39; 13–17 years: *n* = 37) or face-to-face (8–12 years: *n* = 113; 13–17 years: *n* = 111), and parents filled in questionnaires in paper-pencil form. Figure 1 provides an overview on the sample attrition.

The present study focuses on adolescents aged 13 to 17 years and their proxies who filled in the German version of the RPQ.

### 2.2. Ethical Approval

The QOLIBRI-Kid/Ado study was conducted in accordance with all relevant legal regulations including but not limited to the ICH Harmonized Tripartite Guideline for Good Clinical Practice (“ICH GCP”) and the World Medical Association Declaration of Helsinki (“Ethical Principles for Medical Research Involving Human Subjects”). The study attained ethical clearance at each of the recruitment centers and informed consent for all participants, in line with German data protection laws (General Data Protection Regulation, DSGVO). The Ethics Committee of the University Medical Center Göttingen has approved the study (application no.: 19/4/18).

### 2.3. Sociodemographic and Injury-Related Data

Sociodemographic data included the gender and age of the adolescents and the proxies. Injury-related information was collected comprising TBI severity (mild, moderate, severe), presence of lesions visible on cerebral magnetic resonance imaging and/or computed tomography scans (no lesions vs. at least one), and time since injury. Individuals’ functional recovery after TBI was coded based on the clinical rating of pediatric recovery, the Kings Outcome Scale for Childhood Head Injury (KOSCHI) [27] as 1 = ‘dead’, 2 = ‘vegetative state’, 3a = ‘lower severe disability’, 3b = ‘upper severe disability’, 4a = ‘lower moderate disability’, 4b = ‘upper moderate disability’, 5a = ‘good recovery’, or 5b = ‘intact recovery’. Only individuals with values 3 and above were included in the study.

### 2.4. Instruments

#### 2.4.1. Rivermead Post-Concussion Symptoms Questionnaire (RPQ)

The RPQ assesses 16 common PCS which are rated on a five-point Likert-type scale. The total score is calculated as the sum of all item ratings above the value of 1, indicating stronger impairment after TBI, and ranges from 0 (no increased difficulties since TBI) to 64 (most severe symptoms). In adult samples, the instrument has demonstrated good to excellent test–retest reliability (r_tt_ = 0.90) [14], split-half reliability (r = 0.82 to r = 0.95), and internal consistency (Cronbach’s α: 0.89 to 0.93) [23] as well as moderate to high convergent validity and good discriminant validity [23]. The factor structure of the RPQ has been repeatedly examined [28,29,30,31,32,33,34] due to insufficient replication of the original one-factor structure in independent research. While there is still no consensus on the most suitable factorial structure, evidence points toward a three-factor solution including cognitive, emotional, and somatic scales [28] demonstrate a good fit both cross-sectionally (including multiple languages) [24,35] and longitudinally [36]. Therefore, in addition to the conventional RPQ total score, the analyses reported here covered the three scales.

In the present study, the self-report version as well as the newly adapted proxy-assessment form of the RPQ were administered. These two forms of the questionnaire differed solely regarding the wording in the introductory instructions (i.e., “do you suffer from...” or “does your child suffer from...”). Missing values on one to four RPQ items were imputed by prorating the scale mean, while participants with five or more missing values were excluded from further analyses.

#### 2.4.2. Post-Concussion Symptom Inventory (PCSI)

The PCSI [20,21] assesses PCS by asking the individuals after TBI to rate the intensity of symptoms relative to before the injury. The age-adapted PCSI-SR13 Rapid Version [21] comprises 21 items relating to symptom severity before (pre-injury) and after (post-injury) TBI forming three scales (cognitive, emotional, and somatic). The PCSI-SR13 utilizes a seven-point Likert scale (0 = ‘not a problem at all’ to 6 = ‘serious problem’) and the total score ranges from 0 to 132 with higher values indicating more severe PCS. A retrospective-adjusted post-injury difference (RAPID) can be calculated to assess clinically significant changes between the pre- and post-injury status with a change of 80% (either improvement or worsening) considered clinically relevant. Previous research in adolescents (13–22 years) [21] has reported good to excellent internal consistency of the English version of the PCSI-SR13 total score (Cronbach’s α = 0.94) and its scales (Cronbach’s α: 0.79 to 0.93), test–retest reliability (r_tt_ = 0.64 to r_tt_ = 0.79), as well as evidence for convergent, predictive, and factorial validity. In the present study, the proxies filled in the respective PCSI parent-form and only results obtained from the post-injury version of the PCSI were involved in the psychometric analyses with the RPQ.

#### 2.4.3. Generalized Anxiety Disorder Scale 7 (GAD-7)

The GAD-7 [21,37] is a PROM consisting of seven symptoms related to generalized anxiety disorder according to the DSM-IV [38]. Symptoms are rated on a four-point Likert scale (0 = ‘not bothered at all’ to 3 = ‘bothered nearly every day’). The total score ranges from 0 to 21 with higher values indicating greater disturbance, and scores greater than or equal to 5, 10, or 15 representing mild, moderate, or severe impairment, respectively [37]. Previous research in healthy adolescents (10–18 years) has demonstrated good to excellent internal consistency (Cronbach’s α: 0.91 to 0.95), as well as evidence of convergent, and factorial validity [39,40]. Multiple previous studies have administered the GAD-7 to assess anxiety in adolescent TBI samples but lacked an investigation of its psychometric characteristics [41,42].

#### 2.4.4. Patient Health Questionnaire 9 (PHQ-9)

The PHQ-9 [43] is as a nine-item PROM screening for major depression based on the DSM-IV [38] criteria. Items are rated on a four-point Likert scale (0 = ‘not bothered at all’ to 3 = ‘bothered nearly every day’). Total score ranges from 0 to 27 with higher values indicating more severe impairment. Scores between 1 and 4 indicate minimal depression [43,44], and values greater than or equal to 5, 10, or 15 represent mild, moderate, or severe impairment, respectively. Previous research in healthy adolescents (14–18 years) has shown good internal consistency (Cronbach’s α: 0.84) and test–retest reliability (r_tt_ = 0.80 to r_tt_ = 0.88), as well as convergent, criterion, and factorial validity [45,46]. Like the GAD-7, the PHQ-9 has been used in multiple studies to assess depressive symptoms in adolescents after TBI [47,48].

### 2.5. Statistical Analyses

Descriptive statistics were calculated for sociodemographic (adolescents and proxies) and injury-related (adolescents only) characteristics. With certain exceptions (for details, see section on construct validity), all analyses were carried out for both the self-report and the proxy version of the German RPQ.

#### 2.5.1. Item Characteristics

Item characteristics included the absolute and relative frequencies of missing values, mean (M), standard deviation (SD), skewness (SK) and kurtosis (KU). Values ranging from −2 to +2 for SK and KU were considered acceptable [49]. Individuals’ response behavior was analyzed with regard to absolute and relative frequencies in the rating categories. Floor or ceiling effects were considered present if the lowest or highest response categories accounted for more than 15 percent of the responses, respectively [50]. The Shapiro–Wilk normality test examined whether the total and scale scores of the PROMs were normally distributed.

#### 2.5.2. Reliability

The reliability analyses included calculations of Cronbach’s α and McDonald’s ω for the RPQ total and the scales scores, with values between 0.70 and 0.95 (Cronbach’s α) and above 0.80 (McDonald’s ω) indicating good to excellent internal consistency [50,51]. Additionally, Cronbach’s α values after the individual omission of each item were computed to investigate whether the scale would be more consistent after dropping the respective item as indicated if the results exceeded the initial α.

Corrected item-total correlations (CITC) examined the association of the items with the total or scale score. Correlation coefficients of r ≥ 0.30 were considered acceptable [52].

#### 2.5.3. Factorial Validity

A confirmatory factor analysis (CFA) with robust weighted least squares estimators (WLSMV) [53] for ordered categorical data was used to evaluate the latent factor structure of the RPQ. Analyses involved both the original one-factor model [14] and the previously proposed three-factor model encompassing cognitive, emotional, and somatic scales [28]. Models were considered to have a good fit if the cut-off criteria in the following indices (provided in parentheses) were met: Comparative Fit Index (CFI ≥ 0.95), Tucker–Lewis-Index (TLI ≥ 0.95), standardized root mean square residual (SRMR ≤ 0.08), and root mean square error of approximation including 90% confidence interval (RMSEA [CI90%]: excellent fit at 0.05, mediocre fit at 0.10) [54,55]. Additionally, goodness-of-fit was evaluated using the ratio of chi-square to degrees of freedom (*df*), where values < 2 indicate a good fit [56].

#### 2.5.4. Construct Validity

To examine the construct validity of the RPQ in adolescents and proxies, we focused on the convergence of the RPQ with the PCSI and conducted analyses using the GAD-7 and the PHQ-9 for convergent and discriminant validity. Furthermore, we tested hypotheses both regarding emotional states as well as sociodemographic and clinical factors.

Given the non-normal distribution of the measures, correlational validity analyses using Spearman’s ρ were performed between the RPQ and the PCSI-SR13 on both the total and scale score level. Correlation coefficients were evaluated according to Cohen’s conventions as small (|0.10| ≤ ρ < |0.30|), moderate (0.30 ≤ |ρ| < 0.50), or large (ρ ≥ |0.50|) effects [52]. High positive correlations (i.e., ρ ≥ 0.50) between the RPQ and PCSI-SR13 were expected, suggesting that both measures capture the same construct.

Convergent and divergent validity were assessed using Spearman correlations of the RPQ total and scale scores with anxiety and depression, as measured using the GAD-7 and the PHQ-9. We assumed at least moderate positive correlations (ρ ≥ 0.30) as an appropriate convergence between the RPQ emotional scale and anxiety or depression. With regard to the divergent validity of the somatic and cognitive scales in comparison to the GAD-7 and the PHQ-9, values equal to or below 0.30 were considered acceptable.

We examined construct validity by testing hypotheses on concerning gender, TBI severity, and functional recovery using self-reported RPQ data. Higher RPQ values were expected in females compared to males [57], adolescents after moderate and severe TBI compared with those after mild TBI [58], and in individuals who had not fully recovered (KOSCHI < 5b) in contrast to those who showed a complete recovery (KOSCHI = 5b) [59]. In addition, we compared participants with different levels of anxiety and depression (i.e., no or minimal symptom burden: 0–4 vs. at least mild symptom burden: ≥5) using both self-reported and proxy assessed RPQ data. We hypothesized that higher RPQ scores would be associated with higher levels of emotional distress [58,60]. All group comparisons were performed using non-parametric Mann–Whitney-U-tests for independent samples. Cliff’s *δ* determined the effect size for the group comparisons with the following cut-off values: *δ* < |0.28| (small), 0.28 < |*δ*| < 0.43 (medium), and *δ* ≥ |0.43| (large) [61].

### 2.6. Agreement between Self-Report and Proxy Assessment

The concordance of RPQ self-reports with proxy-assessments was analyzed to evaluate the validity of proxy-ratings of PCS after pediatric TBI. Intraclass correlation coefficients (ICC) were calculated for each item and classified as poor (<0.40), fair (0.40–0.59), good (0.60–0.74), or excellent (≥0.75) agreement [62].

All calculations were carried out in R (version 4.1.0, R Foundation for Statistical Computing, Vienna, Austria) [63] using the packages lavaan [64] and psych [65]. The significance level was set at 5% for all analyses except for multiple comparisons of the RPQ scale scores, for which Bonferroni-corrected significance levels were applied.

## 3. Results

### 3.1. Sample Characteristics

The majority of participating adolescents were male (57%), diagnosed with mild TBI (74%), had sustained a TBI four to ten years before study enrollment (59%), and were classified as fully recovered as measured by the KOSCHI (68%). RPQ proxy assessments were most frequently obtained from participants’ mothers (82%). Table 1 provides an overview of the demographic and clinical characteristics of the study sample.

Table 2 provides descriptive statistics on the total and scale scores of the applied PROMs. The average total scores of the RPQ and the PCSI-SR13 were M = 8.17 (SD = 9.96) and M = 16.82 (SD = 15.39) for self-report as well as M = 7.70 (SD = 10.25) and M = 10.38 (SD = 13.55) for proxy assessments, respectively. The means of the GAD-7 and the PHQ-9 total scores were above 3 and 4, irrespective of the source of report. Additionally, participants mostly reported no or minimal anxiety (*n* = 110, 74%) and no or minimal depressive (*n* = 89, 60%) symptoms. The results of the Shapiro–Wilks test indicated a significant deviation from normal distributions for all scales.

### 3.2. Item Characteristics

The RPQ items displayed the following overall characteristics: *M* = 0.76, *SD* = 0.98, *SK* = 1.41, *KU* = 1.70 (self-report) and *M* = 0.66, *SD* = 0.99, *SK* = 1.90, *KU* = 4.34 (proxy assessment) with less than 5% missing values. The distribution of all item scores was skewed to the right with floor effects. For more details, see Appendix A, Table A1.

### 3.3. Reliability

The internal consistency of the RPQ total score as well as of the three scales was good to excellent according to Cronbach’s α and McDonald’s ω values. All RPQ items demonstrated at least satisfactory correlations with the corresponding scale both for self-reports as well as for proxy assessments. However, the omission of one item (Double Vision) in the self-reported RPQ somatic scale yielded a higher Cronbach’s α. In the proxy-assessed RPQ, omission of the items Longer to Think, Double Vision, and Restless led to an increase in Cronbach’s α for the cognitive, somatic, and emotional scales, respectively. Furthermore, omission of the item Double Vision also increased Cronbach’s α for the total scale. For details, see Table 3.

### 3.4. Factorial Validity

CFA results indicated superior fit for the three-factor model as reflected by the fit indices. Direct model comparisons revealed a significantly better fit of the three-factor model in the self-report as well as in the proxy assessment compared to the unidimensional structure. For details, see Table 4.

### 3.5. Convergent and Divergent Validity

Table 5 shows correlations between the RPQ and the PCSI-SR13 total and scale scores for self-reports and proxy assessments. As expected, the total scores as well as all scales were highly correlated (i.e., ρ ≥ 0.50).

The correlations between the GAD-7 and the self-reported RPQ total score (A) as well as with the proxy-assessed RPQ total score (P) were moderate and high, respectively (ρ*_A_ =* 0.36*,* ρ*_p_* = 0.53). As expected, this correlation was mainly due to the strong association between items from the RPQ emotional scale and the GAD-7 score (ρ*_A_* = 0.38, r*_P_* = 0.59). Both in the self-reports and in the proxy assessments, the RPQ somatic and the cognitive scales were moderately associated with GAD-7 scores (0.30 < ρ < 0.50). Thus, higher RPQ scores were associated with higher levels of anxiety.

The correlation of the PHQ-9 with the self-reported RPQ total score was moderate (ρ = 0.35), and the relationship with proxy-assessed RPQ values was strong (ρ = 0.62). As expected, the correlation between the RPQ and the PHQ-9 was mainly driven by the association with the emotional scale (ρ*_A_* = 0.33, ρ*_P_* = 0.57). The RPQ somatic (S) and cognitive (C) scales were moderately associated with depressive symptoms in the self-reports (ρ*_S_* = 0.31, ρ*_C_* = 0.30) and moderately to strongly correlated in the proxy assessments (ρ*_S_* = 0.48, ρ*_C_* = 0.52). Overall, higher RPQ scores were related to higher levels of depression. For details, see Table 6.

Regarding hypothesis testing, group comparisons revealed higher self-reported RPQ total and emotional scores for females than for males. No statistically significant gender difference was observed in the cognitive and somatic scale scores. Further group comparisons revealed statistically significant differences regarding TBI severity in the RPQ cognitive score only. Furthermore, differences in functional recovery were found for the RPQ total score as well as in all scale scores. Finally, evidence of statistically significant group differences for the presence of anxiety as well as depression symptoms was found in the RPQ total and all scale scores. For more details, see Table 7.

### 3.6. Agreement between Self-Report and Proxy Assessment

The item-wise agreement between RPQ self-reports and proxy assessments was fair across all items (ICC [2,1] = 0.44, CI_95%_ [0.41, 0.47]). In contrast, the agreement differed at the scale level. While agreement was fair for the somatic scale (ICC [2,1] =0.49, CI_95%_ [0.45, 0.52]), concordance was poor for the cognitive (ICC [2,1] = 0.38, CI_95%_ [0.32, 0.45]) and the emotional scales (ICC [2,1] = 0.37, CI_95%_ [0.31, 0.43]). Nonetheless, a pronounced overlap was apparent between RPQ self-reports and proxy assessments in the mean ratings of individual items (see Figure 2).

Both in RPQ self-reports (A) and in the proxy assessments (P), the average item scores (see Table A1 in Appendix A) displayed the lowest impairment concerning the items Double Vision (*M_A_* = 0.21, *SD_A_* = 0.59; *M_P_* = 0.12, *SD_P_* = 0.55) and Nausea (*M_A_* = 0.37, *SD_A_* = 0.75; *M_P_* = 0.29, *SD_P_* = 0.73). The largest discrepancies were found regarding the items Poor Concentration (*M_A_* = 1.23, *SD_A_* = 1.18; *M_P_* = 1.14, *SD_P_* = 1.25) and Irritability (*M_A_* = 1.12, *SD_A_* = 1.16; *M_P_* = 1.06, *SD_P_* = 1.18). Relatively large discrepancies between adolescents’ and proxies’ ratings were observed for the items Feeling Depressed (*M_A_* = 0.78, *SD_A_* = 1.18; *M_P_* = 0.51, *SD_P_* = 1.01) and Dizziness (*M_A_* = 0.65, *SD_A_* = 0.88; *M_P_* = 0.46, *SD_P_* = 0.91). Only small discrepancies were found between self-reports and proxy assessments regarding the item Fatigue (*M_A_* = 1.06, *SD_A_* = 1.10; *M_P_* = 0.93, *SD_P_* = 1.16). However, RPQ self-reports overall indicated notable impairment due to fatigue. In contrast, proxies did not report the presence of fatigue symptoms in adolescents after TBI.

## 4. Discussion

The main aim of the present study was to examine the psychometric properties of the German RPQ in adolescents after TBI aged 13–17 years. A secondary target was to compare RPQ self-reports with proxy assessments to evaluate their applicability as a surrogate instrument whenever individuals after TBI are unable to answer for themselves.

Overall, the self-reported RPQ had satisfactory psychometric properties in adolescents after TBI. The results of our study support its validity as a measure of PCS and underline the utility of the RPQ in identifying differences in impairment between sexes, states of functional recovery, and levels of depression and anxiety. Therefore, we conclude that the German version of the RPQ can be administered reliably and validly in adolescents, just as it can be in adults [23]. In addition, our findings demonstrate comparable validity of the proxy-rated RPQ. Although the agreement between self-reports and proxy ratings was fair to poor, we observed a pronounced overlap in item ratings. Therefore, the RPQ self-report should be preferred, while the proxy version may serve as an alternative in cases where the former cannot be administered.

On the item level, all RPQ scores were substantially right-skewed and revealed floor effects both in self-reported as well as proxy-assessed data. This indicates that the symptom burden was rather low in the study sample. Irrespective of the report form, the results yielded satisfactory to excellent item-total correlations and internal consistency. However, the item Double Vision was found to decrease consistency irrespective of the source of report. The same issue has been reported in adults after TBI [23,29], leading to suggestions to remove this item from the instrument due to its poor fit [29]. In the current study, double vision was the least prevalent PCS, which has been reported similarly in other studies with pediatric samples after mild TBI [66,67]. Therefore, the experience of this symptom may be a characteristic either of more acute TBI phases or of more severe TBI cases. Furthermore, previous research has proposed the presence of an underlying ‘vision-related’ factor in RPQ scores comprising the item Double Vision among others [68,69]. In addition to the Double Vision item, omission of the items Longer to Think and Restless led to an increase in Cronbach’s α in the proxy version. Since the present study focused specifically on the comparison of the original unidimensional RPQ structure with the most commonly proposed alternative three-factor model (i.e., cognitive, emotional, somatic), future studies should investigate the model fit of alternative factor solutions in RPQ scores of adolescents after TBI and their proxies.

The conventional one-factor structure as well as the previously suggested three-factor model indicated satisfactory goodness-of-fit. The latter model, however, was found to have a superior fit both for data retrieved from self-reports as well as from proxy assessments. This finding points away from the unidimensional structure of the German RPQ for adolescents and their proxies, which is consistent with the results of studies on the English RPQ and its translations in adults [28,29,30,31,32,33,34]. The comparability of the RPQ scores with concurrent PCS assessments was investigated based on the PCSI-SR13, which is an instrument that was itself adapted from the Post-Concussion Scale [20,70] specifically for the use in children and adolescents. Since the correlation between both tools was high for both the self-report and the proxy assessment, it can be concluded that the RPQ is capable of detecting PCS at least on a comparable level to the PCSI-SR13.

The RPQ scores were moderately to strongly correlated with anxiety and depression, both in self-reports and proxy assessments. Further analysis revealed that these correlations were mainly due to the associations of the RPQ emotional scale with the GAD-7 and PHQ-9. Because of the overlap between PCS, anxiety, and depression [71], differential diagnosis is strongly recommended as a basis for appropriate treatment.

Regarding the hypothesis testing of group differences, female adolescents tended to report statistically significantly higher symptom severity than males in the RPQ total score and the emotional scale. Previous research has reported higher RPQ scores in females both in TBI samples as well as in healthy controls, arguing for normal differences due to a higher prevalence of bodily problems (e.g., back pain) and emotional symptoms (e.g., depression) in women [72]. On the biological level, sex differences in RPQ scores may be mediated by biological mechanisms related to the activity in the hypothalamic-pituitary-adrenal axis as well as by genetic factors [73]. The relationship of gender differences in RPQ scores with sociodemographic and biological predictors in adolescent TBI samples should be targeted by future research.

Comparisons regarding TBI severity revealed that adolescents after moderate and severe TBI reported statistically significantly higher cognitive impairment than those after mild TBI, which is in line with previous literature [58]. However, TBI groups did not differ significantly regarding emotional and somatic symptoms. It must be considered that the majority of participants in the sample had experienced a TBI four to ten years before study enrollment. Therefore, PCS had likely been partially or fully resolved by the time of data collection. The experience of a moderate to severe TBI seems to be related specifically to persistent impairment in the cognitive domain, at least in adults [74]. Therefore, the lack of statistically significant differences between TBI severity states in the RPQ emotional and somatic scales may be explained by distinct recovery processes in different PCS domains. Recovery mechanisms are central for individuals’ disease prognosis as well as the planning of effective treatment and should be studied in more detail in the future, particularly in the field of pediatric TBI.

Individuals with good functional recovery indicated statistically significantly lower RPQ total scores as well as lower scale scores. Previous research has demonstrated a similar relationship between good functional recovery and low PCS in adult TBI populations [23]. Finally, individuals who suffered from depressive and those who experienced anxiety symptoms had statistically significantly higher RPQ total as well as scale scores. This finding is in line with previous reports on the association of PCS with anxiety [75] and depression [76] in adults after TBI. These findings underline that the administration of RPQ self-reports can assist in the identification of functional impairment as well as emotional disturbance following TBI in adolescents and can provide information for the allocation of targeted clinical care.

The overall agreement between RPQ self-reports and proxy assessments was generally fair. However, the concordance was rather poor regarding the cognitive and emotional scales. The discordance between self- and parent assessments has been reported in previous studies in related fields as well. Specifically, the self-reported quality of life of children and adolescents after TBI was found to be statistically significantly lower than ratings derived from proxy judgments [77]. In general populations, parents of adolescents were found to underestimate the emotional distress (i.e., depression, anxiety, anger) of their offspring [78]. Research on cancer survival has also demonstrated limited child–parent agreement with regard to the impression of traumatization after the illness [79]. In contrast, previous studies have found an acceptable agreement between adolescents and parents in their evaluation of psychosocial functioning after pediatric TBI [80]. Research in children and adolescents suffering from chronic pain has highlighted that the highest agreement between self- and parent ratings was achieved for directly observable and salient symptoms (e.g., physical disabilities, family interactions) [81]. This finding is in alignment with the fair adolescent-parent concordance in the RPQ physical scale in the present study. In contrast, Sady and colleagues [21] have found a moderate concordance between PCSI self-report and proxy assessments in pediatric TBI samples. Therefore, the analyses of self-proxy agreement in the present study may have been affected by limitations such as the sample heterogeneity and the relatively small sample size. Furthermore, in a previous longitudinal study, parent–child concordance was found to be higher for children with a higher symptom burden [82]. Consequently, the poor self-proxy agreement in the RPQ cognitive and emotional scales observed in our study may be due to the experience of relatively low-severity PCS and the generally long time since injury. Therefore, the application of the RPQ proxy version may be most appropriate in acute injury phases and/or for adolescents after moderate and severe TBI.

The discordance between self- and proxy reports was perhaps most striking regarding fatigue, with proxies failing overall to notice that adolescents felt fatigued before as well as after TBI. Fatigue is among the most severe PCS [83,84] and underestimating the presence of this symptom may lead to substantial burden for affected adolescents. In fact, general awareness concerning the severity of fatigue has recently increased due to its status as a common outcome of the Post-COVID-19 Syndrome [85]. Further strategies to enhance parents’ as well as health care personnel’s attention, particularly with respect to TBI-related fatigue in children and adolescents, are needed to improve future rehabilitation policy.

Since discrepancies between self- and proxy ratings are affected by characteristics of the individual, the proxy, and the assessed construct [86], the assessment of cognitive and emotional PCS should be based preferably on self-reports. Proxy assessments can however serve as surrogate ratings whenever individuals are unable to answer for themselves. Further studies should focus on improvement of the proxy assessments in order to reduce bias, for example by using statistical methods (e.g., multiple imputation or proxy substitution with adjustment) [87].

### Strengths and Limitations

The current study for the first time systematically investigated the psychometric properties of the RPQ in German-speaking adolescents after TBI. The results presented provide robust evidence for the applicability of this instrument in adolescents after TBI. Careful consideration was given to the administration of the RPQ in adolescents by comparison with the respective standard assessment instrument for PCS (i.e., PCSI-SR13). The inclusion and exclusion criteria were chosen in such a way as to cover a wide range of affected adolescents, which corresponds to the guidelines for psychometric validations [88,89]. Based on these criteria, a study sample was acquired that was sufficient to enable conclusions to be drawn about multiple commonly used indicators of psychometric quality. The factorial validity of the RPQ was investigated for the standard unidimensional as well as the frequently proposed alternative three-factor model of the RPQ. Furthermore, reliability and validity were evaluated for the self-report and proxy assessment versions of the RPQ. We have addressed the direct comparison of self- and proxy-reported outcomes, which is an added value of this study. Literature reviews have shown that most health assessments in pediatric samples after TBI are currently done as proxy ratings [90,91]. To avoid parental response bias, the results of the current study underline the utility of self-reports for PCS assessments in adolescents after TBI. These findings are of crucial importance since parent reports are commonly chosen as the only source of information on the health status in pediatric populations.

The following limitations regarding the data collection process should be noted. First, out of more than 5000 families invited to participate in the study between January 2019 and January 2022, approximately 93% did not participate, which may be indicative of a sample bias. Second, the sample of participants included in the present study showed a pronounced heterogeneity regarding sociodemographic and clinical characteristics. Despite the relatively small sample size, evidence for good psychometric properties was found. However, investigations of potential effects caused by factors such as different severity of TBI were not possible. A closer examination of predictive effects based on data derived from larger surveys is therefore needed in the future. Moreover, analyses of convergent and divergent validity showed moderate to high correlations of PHQ-9 and GAD-7 scores with the RPQ total score, both in self-reports and in proxy assessments, driven particularly by the correlations with the emotional scale. However, depression and anxiety scores also demonstrated moderate to high correlations with somatic and cognitive scales, thus calling into question the discriminatory ability of the PHQ-9 and GAD-7 for PCS assessments. Future studies should investigate the divergent validity of the RPQ with self-report and proxy assessment forms of alternative instruments assessing depression and generalized anxiety in children and adolescents, such as the Children’s Depression Inventory [92], the Penn State Worry Questionnaire [93], and the Child Behavior Checklist for depression or anxiety [94]. Additionally, the proxy assessments were provided mainly by female parents. Since, in fact, health-related proxy ratings are commonly provided by mothers [95], currently under-explored possible gender effects on child–parent concordance should be targeted by future studies. In addition, most adolescents (83%) had suffered a TBI between to 2 and 10 years prior to study enrollment. Consequently, the adolescent-parent concordance on RPQ values in acute recovery phases could not be examined appropriately and remains to be investigated. Finally, as there is no consensus yet as to which cut-off values in RPQ scores robustly indicate clinical relevance of PCS and appropriately capture symptom burden in children and adolescents after TBI, a validation of the cut-off values from previous literature in the field of pediatric TBI [29] is needed.

Future research should aim to establish cut-off values for the evaluation of the clinical relevance of PCS in adolescents. Furthermore, providing reference values for general population samples for the German RPQ as well as other language versions would further increase its applicability in clinical and research contexts. Since the experience of PCS-like symptoms has been reported in general population samples [96,97], it is important to detect the “normal range” of PCS. RPQ reference values in healthy general populations would allow highly impacted individuals in need of targeted clinical interventions to be identified. Moreover, efforts should be made to investigate the relationship between psychosocial and injury-related factors with PCS as measured by the RPQ in samples of sufficient size. In accordance with research conducted in adult TBI samples [98], identifying candidate predictors for the severity of somatic, cognitive, or emotional symptoms (e.g., age, gender, years of education, etc.) could assist in the early therapeutic process and throughout the course of rehabilitation. In addition, more validation studies are needed on the psychometric properties of different RPQ language versions and their validity in pre-teen children. An ongoing study addresses the validation of the RPQ proxy version in children aged 8 to 14 years. Although proxy assessments of health outcomes in children and adolescents have become widely accepted outcome measures [99], the discordance between self-reports and proxy assessments found in the current study encourages the use of the self-reported German RPQ whenever possible, whereas proxies should serve as an alternative only in cases when young patients are unable to provide self-reports.

## 5. Conclusions and Outlook

This study provides the first systematic psychometric evaluation of the German RPQ in adolescents who have experienced a TBI. Our results indicate good to excellent psychometric properties in adolescents and proxies, as well as their comparability with the original and translated versions of the RPQ in adults after TBI [23]. Therefore, we encourage the use of the German RPQ in longitudinal investigations on chronic PCS over the lifespan of individuals after TBI. The findings of previous research suggest that the assessment of chronic PCS based on instruments such as the RPQ should not rely on the total score. Instead, a combination with clinician ratings is recommended [100]. The evidence for the validity of the three-factor model found in the present study is of practical use since the differentiation of cognitive, somatic, and emotional impairments allows the clinical condition of individuals to be described in more detail and treatments to be improved.

## Figures and Tables

**Figure 1 jcm-12-00319-f001:**
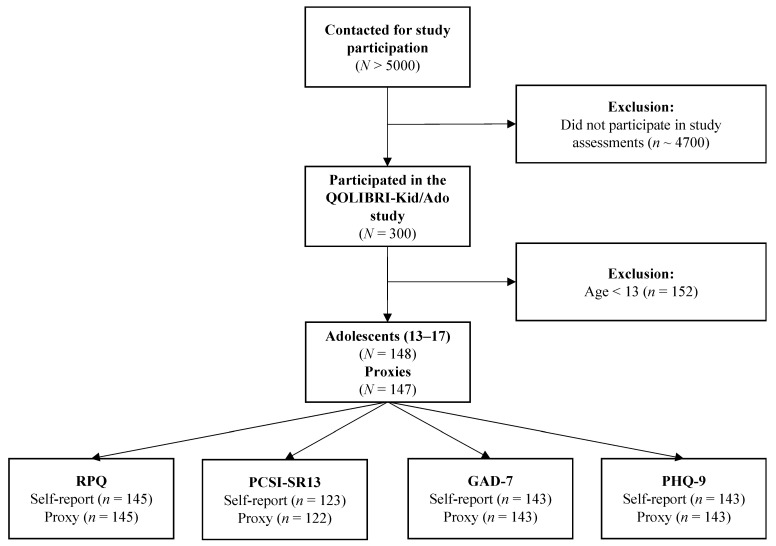
Sample attrition plot.

**Figure 2 jcm-12-00319-f002:**
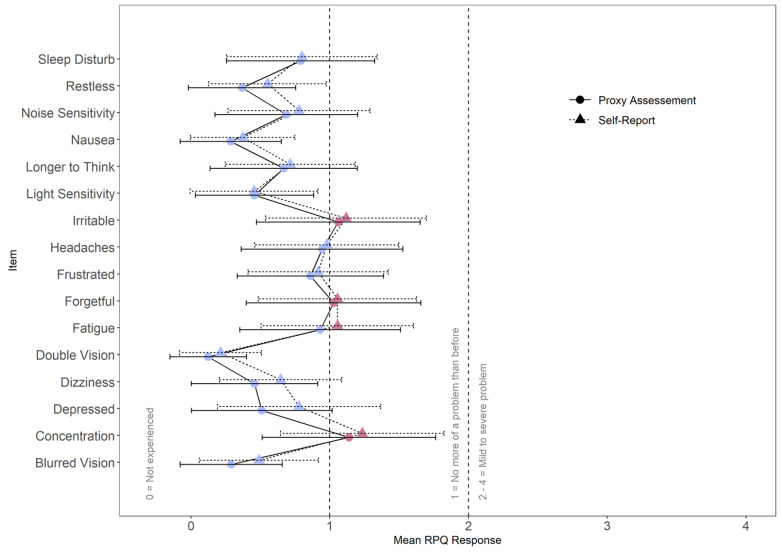
Mean item scores for self-report and proxy assessment. Error bars indicate standard deviations. Red marks indicate mean item values greater than or equal to 1.0.

**Table 1 jcm-12-00319-t001:** Demographic and clinical characteristics of the study sample.

Variable	Group/Values	Adolescents (*n* = 148)	Proxies (*n* =147)
Gender ^†^	Female	62 (42%)	121 (82%)
	Male	85 (57%)	26 (18%)
	Diverse	1 (1%)	0 (0%)
	Missing	0 (0%)	1 (1%)
Age	*M* (*SD*)	15.24 (1.47)	47.97 (5.64)
	*Min–Max*	13.00–17.92	35.00–70.00
TBI severity ^†^	Mild	109 (74%)	-
	Moderate	9 (6%)	-
	Severe	30 (20%)	-
Presence of lesions ^†^	No lesions	100 (68%)	-
	At least one lesion	43 (29%)	-
	Missing	5 (3%)	-
Years since injury ^†^	<1	3 (2%)	-
	1–<2	20 (14%)	-
	2–<4	36 (24%)	-
	4–10	88 (59%)	-
	Missing	1 (13%)	-
KOSCHI Score ^†^	3a (lower severe disability)	0 (0%)	-
	3b	0 (0%)	-
	4a	5 (3%)	-
	4b	18 (12%)	-
	5a	25 (17%)	-
	5b (full recovery)	100 (68%)	-

^†^ For categorical variables, absolute (*n*) and relative (%) frequencies are reported. Note. TBI = traumatic brain injury, KOSCHI = Kings Outcome Scale for Childhood Head Injury, *M* = mean, *SD* = standard deviation, *Min* = minimum, *Max* = maximum.

**Table 2 jcm-12-00319-t002:** Descriptive statistics on the total and scale scores of the applied PROMs.

Source		RPQ				PCSI-SR13					GAD-7	PHQ-9
		C	S	E	Total	C	P	E	F	Total	Total	Total
Self-report	*n*	145	145	145	145	123	123	123	123	123	143	143
	*M*	2.13	3.79	2.25	8.17	5.63	6.18	5.02	3.22	16.82	3.43	4.7
	*SD*	3.03	5.22	3.43	9.96	5.47	6.66	5.52	3.52	15.39	3.57	4.19
	*Mdn*	0	2	0	4	4	4	3	3	12	2	4
	*W*	**0.74**	**0.75**	**0.71**	**0.80**	**0.85**	**0.82**	**0.83**	**0.82**	**0.86**	**0.82**	**0.88**
Proxy assessment	*n*	145	145	145	145	122	122	122	121	122	143	143
	*M*	2.29	3.57	1.84	7.7	3.3	3.75	3.34	2.04	10.38	3.44	4.71
	*SD*	3.29	5.49	3.31	10.25	5.07	5.88	4.8	3.44	13.55	3.58	4.22
	*Mdn*	0	0	0	2	1	2	1	0	6	2	4
	*W*	**0.73**	**0.70**	**0.63**	**0.77**	**0.70**	**0.69**	**0.73**	**0.65**	**0.76**	**0.82**	**0.88**

Note. *n* = number of observations, *M* = mean, *SD* = standard deviation, *Mdn* = median, *W* = value of the Shapiro–Wilk normality test, RPQ = Rivermead Post-Concussion Symptoms Questionnaire, PCSI = Post-Concussion Symptom Inventory, GAD-7 = Generalized Anxiety Disorder Scale 7, PHQ-9 = Patient Health Questionnaire 9. RPQ scales: C = cognitive, S = somatic, E = emotional, Total = total score; PCSI-SR13 scales: C = Cognitive, P = Physical, E = emotional, F = Fatigue, Total = total score; Total = total scores of the GAD-7 and PHQ-9. Values in bold indicate significant deviation from normality (*p* < 0.001).

**Table 3 jcm-12-00319-t003:** Reliability coefficients.

Source	Scale	Cronbach’s α	Cronbach’s α When Item Omitted	McDonald’s ω	Corrected Item-Total Correlation Range	Correlations		
						(1)	(2)	(3)
Self-report	(1) Cognitive	**0.84**	0.74–0.83	**0.84**	0.70–0.81	1	–	–
	(2) Somatic	**0.82**	0.79–0.83	**0.82**	0.34–0.73	**0.57**	1	–
	(3) Emotional	**0.81**	0.72–0.79	**0.82**	0.67–0.80	**0.62**	**0.64**	1
	(4) Total score	**0.90**	0.89–0.90	**0.93**	0.40–0.76	**0.80**	**0.91**	**0.86**
						(1)	(2)	(3)
Proxy assessment	(1) Cognitive	**0.90**	0.82–0.92	**0.91**	0.77–0.91	1	–	–
	(2) Somatic	**0.86**	0.83–0.87	**0.86**	0.37–0.76	**0.56**	1	–
	(3) Emotional	**0.83**	0.71–0.84	**0.84**	0.59–0.89	**0.60**	**0.61**	1
	(4) Total score	**0.91**	0.90–0.92	**0.95**	0.30–0.76	**0.80**	**0.91**	**0.83**

Note. Values in bold indicate at least satisfactory Cronbach’s α (i.e., α ≥ 0.70) and McDonald’s ω (i.e., ω ≥ 0.80) or at least moderate correlation coefficients (i.e., r ≥ 0.30).

**Table 4 jcm-12-00319-t004:** Fit indices and model comparisons of the one- and three-factor models for RPQ self-reports and proxy assessments.

		Confirmatory Factor Analyses (CFA)									Model Comparison	
Source	Model	*χ2*	*df*	*χ2*/*df*	*p*	CFI	TLI	RMSEA	CI_90%_	SRMR	∆*χ2* (∆*df*)	*p*
Self-report	One factor	225.89	104	2.50	*<0.001*	**0.98**	**0.98**	0.09	[0.07, 0.11]	0.1	49.05 (3)	*<0.001*
	Three factors	141.04	101	**1.86**	*0.005*	**0.99**	**1.00**	**0.05**	[**0.03**, 0.07]	0.09	–	*–*
Proxy assessment	One factor	475.72	104	4.14	*<0.001*	**0.96**	**0.96**	0.16	[0.14, 0.17]	0.16	89.68 (3)	*<0.001*
	Three factors	209.72	101	2.40	*<0.001*	**0.99**	**0.99**	0.08	[0.07, 0.10]	0.12	–	–

Note. *χ2* = chi square, *df* = degrees of freedom, *χ2*/*df* = ratio (cut-off: ≤ 2), *p* = *p*-value, CFI = Comparative Fit Index (cut-off: ≥ 0.95), TLI = Tucker–Lewis Index (cut-off: ≥ 0.95), RMSEA = root mean square error of approximation (cut-off: excellent at ≤ 0.05, moderate at ≤ 0.10) with 90% confidence interval (CI), SRMR = standardized root mean square (cut-off: ≤ 0.08), ∆*χ2* = difference in chi-square statistics under Satorra-Bentler correction; ∆*df* = difference in degrees of freedom. Values in bold indicate at least satisfactory model fit according to the respective cut-offs, values in italics are significant at 5%.

**Table 5 jcm-12-00319-t005:** Spearman correlations between the RPQ and PCSI-SR13.

Source	*n*	Total (RPQ, PCSI)	Somatic (RPQ), Physical (PCSI)	Emotional (RPQ, PCSI)	Cognitive (RPQ, PCSI)
Self-report	123	**0.80**	**0.70**	**0.70**	**0.71**
Proxy assessment	121	**0.75**	**0.57**	**0.64**	**0.71**

Note. *n* = absolute frequencies, RPQ = Rivermead Post-Concussion Symptoms Questionnaire, PCSI = Post-Concussion Symptom Inventory. The total PCSI score represents the aggregate of the scales Physical, Emotional, Cognitive, and Fatigue. The Fatigue scale was excluded from the correlation analyses between the RPQ and the PCSI-SR13 since the RPQ contains only a single item measuring fatigue. Values in bold indicate high correlation coefficients (i.e., ρ ≥ 0.50).

**Table 6 jcm-12-00319-t006:** Spearman correlations between the RPQ total and scales scores with the GAD-7 and the PHQ-9.

Source	PROM	Total	Somatic	Emotional	Cognitive
Self-report	GAD-7	*0.36*	*0.34*	*0.38*	*0.25*
	PHQ-9	*0.35*	*0.31*	*0.33*	*0.30*
Proxy assessment	GAD-7	**0.53**	*0.37*	**0.59**	*0.42*
	PHQ-9	**0.62**	*0.48*	**0.57**	**0.52**

Note. GAD-7 = Generalized Anxiety Disorder Scale 7, PHQ-9 = Patient Health Questionnaire 9. Values in italics indicate moderate correlation coefficients (i.e., 0.30 ≤ ρ < 0.50); values in bold indicate high correlation coefficients (i.e., ρ ≥ 0.50).

**Table 7 jcm-12-00319-t007:** Means, effect sizes, and Mann–Whitney U-test results for the comparison of the self-reported RPQ total and scale scores with regard to sociodemographic and injury-related characteristics.

Variable	Scale	Group	*n*	*M*	*Md*	Cliff’s *δ*	*W*	*p*
Sex	Total score	Male	83	6.41	3	−0.19	2044	**0.022**
		Female	61	10.54	8			
	Cognitive	Male	83	1.82	0	−0.13	2214.5	0.077 ^☨^
		Female	61	2.59	0			
	Emotional	Male	83	1.55	0	−0.23	1953	**0.005** ^☨^
		Female	61	3.23	2			
	Somatic	Male	83	3.04	2	−0.15	2156.5	0.055 ^☨^
		Female	61	4.72	2			
TBI severity	Total score	Mild	107	7.80	4	−0.12	1795	0.137
		Moderate—Severe	38	9.18	6			
	Cognitive	Mild	107	1.69	0	−0.28	1473.5	**0.003** ^☨^
		Moderate—Severe	38	3.37	3			
	Emotional	Mild	107	2.24	0	−0.02	1994	0.424 ^☨^
		Moderate—Severe	38	2.26	0			
	Somatic	Mild	107	3.87	2	−0.04	1961	0.368 ^☨^
		Moderate—Severe	38	3.55	2			
KOSCHI score	Total score	5b	123	5.09	4	*−1.02*	1534.5	**<0.001**
		<5b	22	13.15	12.5			
	Cognitive	5b	123	1.48	0	*−1.15*	1703	**0.002** ^☨^
		<5b	22	3.98	3.5			
	Emotional	5b	123	1.26	0	*−0.60*	1718.5	**0.003** ^☨^
		<5b	22	3.06	2.5			
	Somatic	5b	123	2.36	2	*−0.83*	1505.5	**<0.001** ^☨^
		<5b	22	6.11	3.5			
GAD-7	Total score	<5	110	5.57	2	*−1.10*	1034.5	**<0.001**
		≥5	37	15.26	14			
	Cognitive	<5	110	1.45	0	*−0.97*	1120.5	**<0.001** ^☨^
		≥5	37	4.17	5			
	Emotional	<5	110	1.44	0	*−1.00*	1090	**<0.001** ^☨^
		≥5	37	4.54	3			
	Somatic	<5	110	2.68	0	*−0.82*	1180.5	**<0.001** ^☨^
		≥5	37	6.54	5			
PHQ-9	Total score	<5	89	5.45	2	*−0.69*	1619	**<0.001**
		≥5	58	11.82	9			
	Cognitive	<5	89	1.38	0	*−0.65*	1769	**0.001** ^☨^
		≥5	58	3.27	2			
	Emotional	<5	89	1.41	0	*−0.62*	1651	**<0.001** ^☨^
		≥5	58	3.43	2			
	Somatic	<5	89	2.66	0	*−0.51*	1785.5	**0.002** ^☨^
		≥5	58	5.12	4			

^☨^ Bonferroni-adjusted significance level for scale comparisons was 5%/3 = 1.67%. Note. TBI = traumatic brain injury, KOSCHI = Kings Outcome Scale for Childhood Head Injury, GAD-7 = Generalized Anxiety Disorder Scale 7, PHQ-9 = Patient Health Questionnaire 9, *n* = sample size, *M* = mean, *W* = Mann–Whitney U-test statistic, *p* = *p*-value in difference test, Cliff’s δ = effect statistic: *δ* < |0.28| (small), |0.28| < *δ* < |0.43| (medium), and *δ* ≥ |0.43| (large). Values in *italics* indicate large effects. Negative values indicate greater impairment in individuals with less favorable recovery or more severely rated symptoms. *p*-values in bold indicate significant results (either on 5% or on 1.67% α level).

## Data Availability

The data presented in this study are available on request from the corresponding authors. Data are not publicly available due to data protection reasons.

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
