# Peer review of "Psychometric Properties of the German Version of the Rivermead Post-Concussion Symptoms Questionnaire in Adolescents after Traumatic Brain Injury and Their Proxies"

_jcm, 2022, doi:10.3390/jcm12010319_

Round 1

Reviewer 1 Report

ID: jcm-2041158

Title: Psychometric properties of the German version of the Rivermead Post-Concussion Symptoms Questionnaire in adolescents after traumatic brain injury and their proxies.

Thank you for providing a chance to review this manuscript.

Comment: major revision.

Detailed information:

Abstract

Line 48-49, page 1: A total of N = 148 adolescents after TBI and N = 147 proxies filled in the RPQ ------ This sentence is not formulated in a standard way.

The abstract is confusing, like several parts cobbled together in a very disorganized way. We generally divide the abstract into four sections: background/purpose, methods, results, and conclusion, which will make the logic of the abstract clearer and easier for the reader.

Introduction

Line 89-91, page 2: underestimated or overestimated and biased perceptions of symptom severity ------ This statement is so general that I think you need to explain it further.

Line 106-107, page 3: The connection between these two paragraphs is very rigid, and the sudden appearance of (QOLIBRI-Kid/Ado) will confuse the reader.

       The logic of the background and the contextualization of the sentences need to be strengthened. I would suggest one topic per paragraph to provide a clearer statement of existing research, the innovation of this study, the purpose of the study, etc.

Materials and Methods

Line 123-124, page 3: What sampling method was used in this study? What were the inclusion and exclusion criteria? Although the sample size is relatively adequate, I would recommend that you provide the calculation of the sample size.

       Line 124-126, page 3: Is this scale applicable to adolescents? Has it been used before in the adolescent population? If so please cite the psychometric properties. In addition, how do you ensure that adolescents fully understand the meaning of the items correctly?

       Line 157-209, page 4-5: Please show the psychometric properties of these scales. A detailed description of the PRQ can be placed in the background section, and only the most critical elements need to be briefly described in the methods section.

Line 248, page 5: Have you ever done a normality test?

Table

Table 1: 1) Please use the standard three-line table; 2) Change N/A to missing; 3) Please keep three decimals; 4) The table should be aesthetically pleasing in addition to displaying the content, please look more at the display of the table in the top issue and make changes.

Table 2: 1) Use of standardized is a three-line table; 2) Abbreviations appearing in the table need to be written in full in the notes.

These tables are more or less problematic, please check each table according to the following aspects: 1) whether to use the standard three-line table; 2) whether the abbreviations appearing in the table are marked with full names; 3) uniform number of decimal points; 4) there are too many tables, please think carefully whether these tables are necessary to appear, or you can consider putting some of them into the appendix; 5) please think carefully about the layout of the chart, the exchange of rows and columns can sometimes greatly improve the beauty of the table.

Discussion

       1) What are the strengths of this study? 2) The discussion part is not deep enough, and there is not enough of your own thinking. The logic of the essay and the articulation between sentences need to be strengthened.

Conclusions and Outlook

Put the outlook in the discussion section, and shorten the conclusion section, keeping only the most important elements

First, reading more papers from the TOP journals, to learn the formats, expressions, and of great importance—logic, might help a lot before revising. Second, rephrase your sentences to make your expressions clear. Some of your sentences and paragraphs are hard to read and understand. Third, present the main findings with tables and results. Do not let the readers do your job. Last, finding a native English speaker to improve the writing can considerably improve the quality.

Thank you and my best,

Your reviewer

Author Response

Response to Reviewer 1 Comments
Title: Psychometric properties of the German version of the Rivermead Post-Concussion Symptoms
Questionnaire in adolescents after traumatic brain injury and their proxies.
Thank you for providing a chance to review this manuscript.
Comment: major revision.
General Response: Dear Reviewer, thank you for your comments and suggestions. We have tried to revise the manuscript according to your recommendations. In addition, we have had the manuscript proofread by a native English speaker. Some of your comments refer to the format of the manuscript, and some of its elements (e.g., tables) point to standards for presenting scientific results. We have done our best to follow your advice. However, we would like to point out that some of your suggestions cannot be adopted because they are related to the guidelines of the Journal of Clinical Medicine. Thank you for your understanding.
Below you will find detailed answers to your questions, which are also highlighted in the manuscript.
Detailed information:
Abstract
Point 1, Line 48-49, page 1: A total of N = 148 adolescents after TBI and N = 147 proxies filled in the RPQ ------ This sentence is not formulated in a standard way.
Response 1: Thank you for this comment. We chose not to report the aggregated total sample size (i.e., N = 296 adolescents and proxies) so as to highlight the difference between the sources of information (i.e., self-report vs. proxy assessment). We have complemented this information with an attrition plot of participants.

To make the sentence more comprehensible, we have changed the wording to: “In total, 148 adolescents after TBI and 147 proxies completed the RPQ.”
For more transparency in terms of sample composition, we have added a sample attrition plot (please, see new Figure 1).

Point 2: The abstract is confusing, like several parts cobbled together in a very disorganized way. We generally divide the abstract into four sections: background/purpose, methods, results, and conclusion, which will make the logic of the abstract clearer and easier for the reader.
Response 2: Thank you for this comment. We have followed the guidelines of the Journal of Clinical Medicine stating the following:
“Abstract. A single paragraph of about 200 words maximum. For research articles, abstracts should give a pertinent overview of the work. We strongly encourage authors to use the following style of structured abstracts, but without headings: (1) Background: Place the question addressed in a broad context and highlight the purpose of the study; (2) Methods: briefly describe the main methods or treatments applied; (3) Results: summarize the article’s main findings; (4) Conclusions: indicate the main conclusions or interpretations.“ (source: Word template JCM; cf. https://www.mdpi.com/journal/jcm/instructions, last access on 18.11.2022)
We have followed the structure of the journal (introduction – method – results – discussion/conclusion) in the design of the abstract and, therefore, did not use headings and did not present the content in one paragraph:
“Background/Aim: The Rivermead Post-Concussion Symptoms Questionnaire (RPQ) is frequently used to assess post-concussion symptoms (PCS) in adults after traumatic brain injury (TBI), showing good to excellent psychometric properties. The current study addresses the lack of empirical evidence in individuals after pediatric TBI, aiming to investigate the psychometric characteristics of self-report and proxy versions of the German RPQ in adolescents (13–17 years) after TBI.
Methods: The psychometric evaluation comprised analyses of reliability and validity on the total and scale (i.e., cognitive, somatic, and emotional) score level. Construct validity was investigated via correlations with the Post-Concussion Symptoms Inventory (PCSI-SR13), Generalized Anxiety Disorder Scale 7 (GAD-7) and Patient Health Questionnaire 9 (PHQ-9), followed by comparisons of RPQ scores regarding sociodemographic and injury characteristics. Adolescent-proxy agreement was assessed through intraclass correlation coefficients (ICC).
Results: In total, 148 adolescents after TBI and 147 proxies completed the RPQ. Cronbach’s α ranged from 0.81 to 0.91. The three-factor structure outperformed the original unidimensional model in the selfreport and proxy versions. The RPQ was strongly correlated with the PCSI-SR13 (self-report: r = 0.86, proxy: r = 0.78). Correlations of the RPQ with the GAD-7 and PHQ-9 were moderate to high for selfreports (r = 0.50, r = 0.45) and high for proxy assessments (r = 0.55, r = 0.63). The item-wise agreement between adolescents and proxies was fair on the total score level (ICC[2,1] = 0.44, CI95% [0.41, 0.47]).
Conclusions: Overall, the self-report as well as the proxy version of the German RPQ displayed good to excellent psychometric properties. They are therefore suitable for PCS assessments in adolescents after TBI. As proxy ratings tend to underestimate PCS (e.g., cognitive and emotional symptoms), selfreports are preferable for evaluations. Only if a patient is unable to answer, should a proxy be used as a surrogate.”

Introduction
Point 3, Line 89-91, page 2: underestimated or overestimated and biased perceptions of symptom severity ------ This statement is so general that I think you need to explain it further.
Response 3: Thank you for this comment. We have revised the introduction according to your comments, which among other things led to the omission of this sentence.

Point 4, Line 106-107, page 3: The connection between these two paragraphs is very rigid, and the sudden appearance of (QOLIBRI-Kid/Ado) will confuse the reader.
Response 4: Thank you for this comment. We agree that mentioning the study framework is inappropriate at this stage. We have moved this paragraph to the methods section and revised the objectives accordingly.

Page 3, lines 131-138:
“To date, there have not been any systematic investigations of the psychometric properties of either the original English version or other translations of the RPQ in pediatric samples. The aim of the present study is, therefore, to investigate the classical psychometric properties of the self-report and proxy versions of the German RPQ for the assessment of PCS in adolescents after TBI aged 13–17 years. The respective analyses include examinations of reliability, factorial validity, and construct validity. Evidence of acceptable psychometric properties would suggest the reliable and valid assessment of the PCS in adolescents after TBI.”

Point 5: The logic of the background and the contextualization of the sentences need to be strengthened. I would suggest one topic per paragraph to provide a clearer statement of existing research, the innovation of this study, the purpose of the study, etc.
Response 5: Thank you for this advice. The introduction tried to adhere to the following structure:
1. Existing research concerning:
a. Pediatric traumatic brain injury (TBI) and its sequelae
b. Post-concussion symptoms (PCS)
c. Assessment of PCS (RPQ and PCSI)
2. Innovation of the study: Reasons for (and in favor of) the RPQ validation
3. Aims of the study
We have revised the introduction according to your suggestions (e.g., by adding a more detailed description of the RPQ) trying to improve contextualization of the sentences within the respective paragraphs.
Introduction:
“Traumatic brain injury (TBI) has been identified as the primary injury-related cause of death and disability in children and adolescents worldwide [1]. Reports of the global incidence of pediatric TBI range from 47 to 280 per 100.000 children/adolescents but vary greatly across countries and injury
causes [2], with road traffic accidents, incidental falls, and other accidents being the most common causes of injury in Europe [3]. Pediatric TBI is associated with profound short- and long-term functional[4], cognitive [5], as well as developmental and behavioral [6] impairments. Consequently, pediatric TBI
poses a substantial burden on injury victims, caregivers, as well as global societies at large [7].
Post-concussion symptoms (PCS) are one of the most frequent sequelae after pediatric TBI. They include headaches, pain in various areas of the body, nausea, light- and noise-sensitivity, double vision, trouble focusing, as well as fatigue among others [8]. The prevalence of PCS in pediatric population ranges widely (from 24.5% to 52%) depending on the diagnostic criteria and the time since injury [9]. Most acute PCS are resolved within a few months after injury [10], however chronic symptoms may be observed over a number of years post TBI [11]. Children and adolescents with persistent PCS report lower health-related quality of life (HRQOL) compared with those who recovered fully after injury,
whereby such deficits remain observable after PCS have resolved [12].
PCS screenings are routinely based on self-report instruments that do not require advanced technical equipment or in-depth practice for their administration [13]. One of the most commonly used patientreported outcome measures (PROM) is the Rivermead Post-Concussion Symptoms Questionnaire
(RPQ) [14]. The RPQ covers a total of 16 items (headaches, dizziness, nausea and/or vomiting, noise sensitivity, sleep disturbance, fatigue, irritability, depression, frustration, forgetfulness and poor memory, poor concentration, slow thinking, blurred vision, light sensitivity, double vision, and restlessness). The presence and the severity of these PCS is rated relative to the individuals’ subjective condition before TBI based on a five-point Likert scale (0 = ‘not present at all’, 1 = ‘no more of a problem than before’, 2 = ‘a mild problem’, 3 = ‘a moderate problem’, to 4 = ‘severe problem’). The original English version designed for adults after TBI demonstrated good psychometric properties and has been proposed to be reliable both as a self-report or a clinician-administered measure [14]. In the AngloSaxon context, previous studies have predominantly focused on young, concussed athletes (14–20 years) and provided only limited information on the reliability and the validity of the instrument [15–18]. The authors have reported good reliability (Cronbach’s α = 0.89; split-half reliability r = 0.88), low testretest reliability (rtt = 0.23 to rtt = 0.24) [15], and good discriminative ability between concussed
individuals and controls (Cohen’s d = 1.57) [16]. However, despite its repeated application after pediatric TBI [17,18], the RPQ is listed in the Common Data Elements (CDE) recommendations as a supplemental measure only for use in adults [19].
For PCS assessment in children and adolescents, the CDE recommendations [19] propose the use of the Post-Concussion Symptom Inventory (PCSI) [20,21]. The PCSI is available in age-adjusted pre-TBI and post-TBI forms for children (aged 8–12 years; PCSI-SR8) and adolescents (aged 13–17 years; PCSI-SR13) as well as in a proxy version (e.g., filled in by parents or caregivers) [21]. The understandability of the PCSI forms was investigated in the respective age groups using cognitive debriefing [21]. The PCSI-SR13 is comparable to the RPQ in terms of item content and wording, with the main difference being the form of administration. While the PCSI-SR13 self-report and proxy versions have two separate forms assessing the experience of PCS before and after TBI, the response scale rating of the RPQ comprises both pieces of information jointly.
The evaluation of the psychometric properties of the RPQ in adolescents and their proxies is crucial for its application in lifelong clinical follow-up and multicenter pediatric studies for several reasons. First, the repeated application of the RPQ in pediatric studies has resulted in reports of reasonable prevalence
rates of acute PCS following mild TBI based on self-reports [22] as well as combined proxy versions [23]. Hence, the validation of the RPQ would further strengthen the reliability of these previous results for conclusions in the clinical context. Second, good to excellent psychometric properties of the instrument have been demonstrated across several translations in adults after TBI – including the German version [24,25]. As these findings provide solid evidence in favor of the use of the RPQ in multinational research and clinical practice, further validation would suggest a similarly widespread applicability of this instrument in adolescents. Third, the administration of a single RPQ form covering both pre- and post-TBI symptoms allows for the time-efficient and low-cost assessment of PCS, thereby enabling and simplifying longitudinal PCS assessment across the lifespan. The RPQ for adolescents would then be applicable during an important period in an individual’s life. Finally, the investigation of the validity of proxy assessments can provide deeper insights into the usability of the RPQ as an observer rating of PCS in adolescents after TBI. Proxy assessments could be used as a surrogate for self-reports, whenever the adolescents themselves are not able to report.
To date, there have not been any systematic investigations of the psychometric properties of either the original English version or other translations of the RPQ in pediatric samples. The aim of the present study is, therefore, to investigate the classical psychometric properties of the self-report and proxy versions of the German RPQ for the assessment of PCS in adolescents after TBI aged 13–17 years. The respective analyses include examinations of reliability, factorial validity, and construct validity. Evidence of acceptable psychometric properties would suggest the reliable and valid assessment of the PCS in adolescents after TBI.”

Materials and Methods
Point 6, Line 123-124, page 3: What sampling method was used in this study? What were the inclusion and exclusion criteria?
Response 6: We have provided additional information on the sampling method:
Pages 3-4, lines 141-157:
“The present study was conducted as part of the Quality of Life after Brain Injury for Kids and Adolescents (QOLIBRI-Kid/Ado) project, which aimed to develop the first disease-specific, age-adapted HRQOL instrument for children (8–12) and adolescents (13–17 years) after TBI. The study sample was a retrospective, clinical convenience sample. Participants were recruited from twelve medical centers and hospitals in Germany between January 2019 and January 2022 based on a procedure consisting of multiple steps. Recruitment sites provided patient lists which included pediatric patients with a diagnosis of TBI (ICD code S06*) of the past ten years. Potential participants were selected and invited
by mail for study participation based on inclusion and exclusion criteria. After they had given their informed consent, appointments were arranged for online or in-person assessments in the centers. The inclusion criteria were an age of 8–17 years, the date of TBI diagnosis being at least three months but not more than ten years before study enrollment, available information on the TBI severity (either based on the Glasgow Coma Scale [GCS] [26] or medical reports), as well as the ability to understand and fill in questionnaires. Children and adolescents who suffered from severe epilepsy, severe polytrauma, serious mental illness prior to TBI, or disease leading to death were excluded from this study.”

Point 7: Although the sample size is relatively adequate, I would recommend that you provide the calculation of the sample size.
Response 7: Thank you for this comment. The sample size estimates were done for the QOLIBRIKid/Ado project based on the minimum required sample size for exploratory and confirmatory factor analysis (Mundfrom, Shaw & Ke, 2005) of the newly developed QOLIBRI-Kid/Ado instrument. According to the simulation study by Mundfrom et al. (2005), the sample size should be around 140 participants to have enough power for model estimations. Therefore, at least 140 participants had to be recruited. Since the QOLIBRI-Kid/Ado was the instrument with the highest number of parameters being estimated, we can be sure that this number of observations is sufficient for the validation of the RPQ.
We have added the following information in method section:
Page 4, lines 158-161:
“In total, more than 5000 families were contacted. Based on sample size estimates within the QOLIBRI-Kid/Ado study, at least 140 participants and their proxies had to be included for each age group [27]. In the end, N = 300 subjects (8–12 years: n = 152; 13–17 years: n = 148) were recruited as participants.”

Point 8, Line 124-126, page 3: Is this scale applicable to adolescents? Has it been used before in the adolescent population? If so please cite the psychometric properties.
Response 8: The aim of the present study was to investigate whether the RPQ is applicable in adolescents in the German language. In the Anglo-Saxon context, previous studies have focused predominantly on young, concussed athletes and provided only limited information on the reliability and the validity of the instrument (e.g., Iverson & Gaetz, 2004; Wilde et al., 2008; Gioia et al., 2009).
This is the first study to systematically evaluate the psychometric properties of the (German) RPQ in an adolescent TBI sample. In order to provide a more detailed overview of previous results on the applicability of the RPQ in adolescents, we have added the following information to the RPQ’s description in the introduction:
Page 2, lines 93-101:
“In the Anglo-Saxon context, previous studies have predominantly focused on young, concussed athletes (14–20 years) and provided only limited information on the reliability and the validity of the instrument [15–18]. The authors have reported good reliability (Cronbach’s α = 0.89; split-half reliability r = 0.88), low test-retest reliability (rtt = 0.23 to rtt = 0.24) [15], and good discriminative ability between concussed individuals and controls (Cohen’s d = 1.57) [16]. However, despite its repeated application after pediatric TBI [17,18], the RPQ is listed in the Common Data Elements (CDE) recommendations as a supplemental measure only for use in adults [19].”

Point 9: In addition, how do you ensure that adolescents fully understand the meaning of the items correctly?
Response 9: The item wording of the RPQ corresponds to that of the PCSI-SR13, which was designed specifically to measure post-concussion symptoms in adolescents aged 13 years and above, and cognitively debriefed. Therefore, we can conclude that adolescents understand the items correctly. To further clarify this point, we have added the following information to the manuscript:
Page 3, lines 103-111:
“The PCSI is available in age-adjusted pre-TBI and post-TBI forms for children (aged 8–12 years; PCSISR8) and adolescents (aged 13–17 years; PCSI-SR13) as well as in a proxy version (e.g., filled in by parents or caregivers) [21]. The understandability of the PCSI forms was investigated in the respective
age groups using cognitive debriefing [21]. The PCSI-SR13 is comparable to the RPQ in terms of item content and wording, with the main difference being the form of administration. While the PCSI-SR13 self-report and proxy versions have two separate forms assessing the experience of PCS before and after TBI, the response scale rating of the RPQ comprises both pieces of information jointly.”

Point 10, Line 157-209, page 4-5: Please show the psychometric properties of these scales.

Response 10: We have added information on the psychometric properties of the instruments used in the present study. When information on the use of an instrument in pediatric TBI populations was missing, we included findings on the psychometric properties in pediatric non-TBI populations.
Page 5, lines 226 ff.: PCSI:
“Previous research in adolescents (13–22 years) has reported good to excellent internal consistency of the English version of the PCSI-SR13 total score (Cronbach’s α = 0.94) and its scales (Cronbach’s α: 0.79 to 0.93), test-retest reliability (rtt = 0.64 to rtt = 0.79), as well as evidence for convergent, predictive,
and factorial validity [21].”
GAD-7:
“Previous research in healthy adolescents (10–18 years) has demonstrated good to excellent internal consistency (Cronbach’s α: 0.91 to 0.95), as well as evidence of convergent, and factorial validity [41,42]. Multiple previous studies have administered the GAD-7 to assess anxiety in adolescent TBI samples but lacked an investigation of its psychometric characteristics [43,44].”
PHQ-9:
“Previous research in healthy adolescents (14–18 years) has shown good internal consistency (Cronbach’s α: 0.84) and test-retest reliability (rtt = 0.80 to rtt = 0.88), as well as convergent, criterion, and factorial validity [47,48]. Like the GAD-7, the PHQ-9 has been used in multiple studies to assess depressive symptoms in adolescents after TBI [49,50].”

Point 11: A detailed description of the PRQ can be placed in the background section, and only the most critical elements need to be briefly described in the methods section.
Response 11: Thank you for this comment. We have moved the main RPQ description including information on the application of the RPQ in adolescents to the introduction section and provided additional information on previously reported psychometric properties of the RPQ in adults (including contradictory findings on its factorial structure) in the method section.
Page 2, lines 82-101:
“One of the most commonly used patient-reported outcome measures (PROM) is the Rivermead PostConcussion Symptoms Questionnaire (RPQ) [14]. The RPQ covers a total of 16 items (headaches, dizziness, nausea and/or vomiting, noise sensitivity, sleep disturbance, fatigue, irritability, depression, frustration, forgetfulness and poor memory, poor concentration, slow thinking, blurred vision, light sensitivity, double vision, and restlessness). The presence and the severity of these PCS is rated relative to the individuals’ subjective condition before TBI based on a five-point Likert scale (0 = ‘not present at all’, 1 = ‘no more of a problem than before’, 2 = ‘a mild problem’, 3 = ‘a moderate problem’,
to 4 = ‘severe problem’). The original English version designed for adults after TBI demonstrated good psychometric properties and has been proposed to be reliable both as a self-report or a clinicianadministered measure [14]. In the Anglo-Saxon context, previous studies have predominantly focused on young, concussed athletes (14–20 years) and provided only limited information on the reliability and the validity of the instrument [15–18]. The authors have reported good reliability (Cronbach’s α = 0.89; split-half reliability r = 0.88), low test-retest reliability (rtt = 0.23 to rtt = 0.24) [15], and good discriminative ability between concussed individuals and controls (Cohen’s d = 1.57) [16]. However, despite its repeated application after pediatric TBI [17,18], the RPQ is listed in the Common Data Elements (CDE) recommendations as a supplemental measure only for use in adults [19].”
Page 5, lines 193-214:
“The RPQ assesses 16 common PCS which are rated on a five-point Likert-type scale. The total score is calculated as the sum of all item ratings above the value of 1, indicating stronger impairment after TBI, and ranges from 0 (no increased difficulties since TBI) to 64 (most severe symptoms). A cut-off of 12 has been suggested for use in adults for clinical screening [29]. Previous research in international adult TBI samples has demonstrated good to excellent test-retest reliability (rtt = 0.90) [14], split-half reliability (r = 0.82 to r = 0.95), and internal consistency (Cronbach’s α: 0.89 to 0.93) [24]. In addition, evidence points towards moderate to high convergent validity and good discriminant validity [24]. The factor structure of the RPQ has been repeatedly examined in adult TBI-populations [29–35] due to insufficient replication of the original one-factor structure in independent research. While there is still no consensus on the most suitable factorial structure, evidence points away from a unidimensional solution. Recent studies in adults after TBI have found that the three-factor solution including cognitive, emotional, and somatic scales [36] demonstrate a good fit both cross-sectionally (including multiple languages) [25,37] and longitudinally [38]. Therefore, in addition to the conventional RPQ total score, the analyses reported here covered the three scales.
In the present study, the self-report version as well as the newly adapted proxy-assessment form of the RPQ were administered. The two forms of the questionnaire differed solely regarding the wording in the introductory instructions (i.e., “do you suffer from...” or “does your child suffer from...”). Missing values on one to four RPQ items were imputed by prorating the scale mean, while cases with five or more missing values were excluded from further analyses.”

Point 12, Line 248, page 5: Have you ever done a normality test?
Response 12: Following your comment, we calculated Shapiro-Wilk normality tests for the scores of all PROMs and found them to deviate significantly from a normal distribution. For this reason, and to ensure the robustness of the results, we calculated Spearman rank correlations and applied MannWhitney U tests for independent samples. The procedure is noted accordingly in the methods and
results sections.
Page 6, lines 269-270:
“The Shapiro-Wilk normality test examined whether the total and scale scores of the PROMs were normally distributed.”
Page 7, lines 297-321:
“Given the non-normal distribution of the measures, correlational validity analyses using Spearman’s ρ were performed between the RPQ and the PCSI-SR13 on both the total and scale score level. Correlation coefficients were evaluated according to Cohen’s conventions as small (0.10 ≤ |ρ| < 0.30), moderate (0.30 ≤ |ρ| < 0.50), or large (ρ ≥ |0.50|) effects [53]. High positive correlations (i.e., ρ ≥ 0.50) between the RPQ and PCSI-SR13 were expected, suggesting that both measures capture the same construct.
Convergent and divergent validity were assessed using Spearman correlations of the RPQ total and scale scores with anxiety and depression, as measured using the GAD-7 and the PHQ-9. We assumed at least moderate positive correlations (ρ ≥ 0.30) as an appropriate convergence between the RPQ emotional scale and anxiety or depression. With regard to the divergent validity of the somatic and cognitive scales in comparison to the GAD-7 and the PHQ-9, values equal to or below 0.30 were considered acceptable.
We investigated construct validity by testing hypotheses on gender differences, as well as differences between TBI severity groups (mild vs. moderate/severe) and recovery after TBI (fully recovered [KOSCHI = 5b] vs. not fully recovered [KOSCHI < 5b]) using self-reported RPQ data. Higher RPQ values were expected in females compared to males [58], adolescents after moderate and severe TBI
compared with those after mild TBI [59], and in individuals who had not fully recovered in contrast to those who had a complete recovery [60]. In addition, adolescents and proxies who reported no or minimal anxiety or depression (scores of 0 to 4) were compared with those who reported at least mild anxiety or depression (scores of ≥ 5) We hypothesized that higher RPQ scores would be associated with higher levels of emotional distress [59,61]. All group comparisons were performed using nonparametric Mann-Whitney-U-tests for independent samples. Cliff’s δ determined the effect size for the group comparisons with the following cut-off values: δ < |0.28| (small), 0.28 < |δ| < 0.43 (medium), and δ ≥ |0.43| (large) [62].”

Table
Point 13, Table 1: 1) Please use the standard three-line table; 2) Change N/A to missing; 3) Please keep three decimals; 4) The table should be aesthetically pleasing in addition to displaying the content, please look more at the display of the table in the top issue and make changes.
Response 13:
1. We have used the template of the JCM suggesting the applied rules between the entries (e.g., between Gender and Age).
2. Revised.
3. We would like to kindly decline this suggestion, since the formatting guidelines (e.g., APA) suggest using two decimal places for all numbers except for p-values (three decimals)
4. As mentioned in the first point, we have presented the results according to the journal guidelines. Final formatting is performed by the journal team, which would resize the tables and place them in the most suitable locations.

Point 14, Table 2: 1) Use of standardized is a three-line table; 2) Abbreviations appearing in the table need to be written in full in the notes.
Response 14:
1. As mentioned above, we have used the template of the JCM suggesting using rules between the entries.
2. Revised

Point 15: These tables are more or less problematic, please check each table according to the following aspects: 1) whether to use the standard three-line table; 2) whether the abbreviations appearing in the table are marked with full names; 3) uniform number of decimal points; 4) there are too many tables, please think carefully whether these tables are necessary to appear, or you can consider putting some of them into the appendix; 5) please think carefully about the layout of the chart, the exchange of rows and columns can sometimes greatly improve the beauty of the table.

Response 15:
1. We are afraid that we cannot change the formatting since the tables correspond to the template of the JCM.
2. We have revised all tables and have added all missing abbreviations.
3. We used two decimal points, except for p-values (which were reported using three decimal points), in accordance with most guidelines for reporting metrics in scientific studies (e.g., APA).
4. We believe that all tables are essential to show the results of the RPQ’s validation. Therefore, we would prefer to keep them in the main text instead of moving them to the appendix. To improve the flow of the text, we have revised Table 2 (presenting source of report as rows, not as columns, to achieve more congruence with other tables) and collapsed Tables 7–12 into one.
5. We have revised Figure 1 according to your suggestions and found that swapping the axes has improved the readability/aesthetics of the figure.

Discussion
Point 16: 1) What are the strengths of this study? 2) The discussion part is not deep enough, and there is not enough of your own thinking. The logic of the essay and the articulation between sentences need to be strengthened.

Response 16: The aim of the study was to evaluate the psychometric properties of the German version of the RPQ and its applicability among adolescents and their proxies. Based on the results of the analyses, we were able to draw conclusions about its reliability and validity. In addition, we compared self- and proxy assessments to determine whether parents or caregivers are able to assess adolescents' symptom burden after a TBI. As this is a psychometric validation study, all analyses were based on guidelines for such validation studies. Interpretations were based on cut-offs and significance tests of conducted statistical analyses. The results were embedded in the current research context, and limitations were discussed.
We have added the following summary of the study strengths:
Page 16, lines 589-607:
“The current study for the first time systematically investigated the psychometric properties of the RPQ in German-speaking adolescents after TBI. The results presented provide robust evidence for the applicability of this instrument in adolescents after TBI. Careful consideration was given to the administration of the RPQ in adolescents by comparison with the respective standard assessment
instrument for PCS (i.e., PCSI-SR13). The inclusion and exclusion criteria were chosen in such a way as to cover a wide range of affected adolescents, which corresponds to the guidelines for psychometric validations [89,90]. Based on these criteria, a study sample was acquired that was sufficient to enable conclusions to be drawn about multiple commonly used indicators of psychometric quality. The factorial validity of the RPQ was investigated for the standard unidimensional as well as the frequently proposed alternative three-factor model of the RPQ. Furthermore, reliability and validity were evaluated for the self-report and proxy assessment versions of the RPQ. We have addressed the direct comparison of self- and proxy-reported outcomes, which is an added value of this study. Literature reviews have shown that most health assessments in pediatric samples after TBI are currently done as proxy ratings [91,92].
To avoid parental response bias, the results of the current study underline the utility of self-reports for PCS assessments in adolescents after TBI. These findings are of crucial importance since parent reports are commonly chosen as the only source of information on the health status in pediatric populations.”

Conclusions and Outlook
Point 17: Put the outlook in the discussion section, and shorten the conclusion section, keeping only the most important elements.

Response 17: Thank you, we have moved the outlook to the discussion section, which resulted in a shortening of the conclusion.
Page 17, lines 659-670:
“This study provides the first systematic psychometric evaluation of the German RPQ in adolescents who have experienced a TBI. Our results indicate good to excellent psychometric properties in adolescents and proxies, as well as their comparability with the original and translated versions of the RPQ in adults after TBI [24]. Therefore, we encourage the use of the German RPQ in longitudinal
investigations on chronic PCS over the lifespan of individuals after TBI. The findings of previous research suggest that the assessment of chronic PCS based on instruments such as the RPQ should not rely on the total score. Instead, a combination with clinician ratings is recommended [101]. The evidence for the validity of the three-factor model found in the present study is of practical use since the differentiation of cognitive, somatic, and emotional impairments allows the clinical condition of individuals to be described in more detail and treatments to be improved.”

Point 18: First, reading more papers from the TOP journals, to learn the formats, expressions, and of great importance—logic, might help a lot before revising. Second, rephrase your sentences to make your expressions clear. Some of your sentences and paragraphs are hard to read and understand. Third, present the main findings with tables and results. Do not let the readers do your job. Last, finding a native English speaker to improve the writing can considerably improve the quality.

Response 18: Thank you for all your efforts to improve our manuscript. First, we had to follow the journal’s guidelines concerning style and formatting and could, therefore, not integrate all your layout suggestions. Second, in psychometric studies, this kind of presentation of results is very common (e.g.,
Table 3 or 4, which cannot be converted into a classic three-line table). Third, we have tried to revise our sentences/paragraphs according to your suggestions and had the manuscript proofread by a native speaker. Finally, we hope to have included the most important information about the RPQ validation results and made it easier for readers to understand our tables by highlighting them (e.g., using bold or italic font).

Thank you and my best,
Your reviewer
Thank you!

Reviewer 2 Report

Dear authors, I have only one minor comment: please explain why you validated the scale only for adolescents. Is it too complex for children?

Round 2

Reviewer 1 Report

ID: jcm-2041158

Title: Psychometric properties of the German version of the Rivermead Post-Concussion Symptoms Questionnaire in adolescents after traumatic brain injury and their proxies.

I appreciate your efforts to improve the manuscript and to respond to the comments made in the first review process. However, there is still one minor issue that needs to be addressed:

Detailed information:

Abstract

Line 38-58, page 1-2: The guidelines require abstracts to be no more than 200 words, and I think this abstract is well over the limit.

Introduction

Line 96-101, page 2: It might be a better choice to put these more detailed contents in the discussion section.

Materials and Methods

       Line 201-214, page 5: I suggest moving this section to the discussion section.

Line 234, 246, page 6: The abbreviations “GAD-7”, “PHQ-9” already appear in the subtitle and can be used directly.

Line 192-256, page 5-6: I recommend reducing the sentences in this section and trying to use more concise words to express more rich content.

Line 271-277, page 6: Please add the reliability indicator “MacDonald’s ω”.

Line 309-321, page 7: This paragraph is not clear about the hypothesis, please reorganize the logic and rewrite.

Table 4

       “χ2” and “df should be italicized, please check if the same error occurs in the full text.

Figure 2

       This image is very blurry. To give the reader a better sensory experience, please try using a higher resolution image.

Thank you and my best,

Your reviewer

Author Response

Response to Reviewer 1 Comments

Title: Psychometric properties of the German version of the Rivermead Post-Concussion Symptoms Questionnaire in adolescents after traumatic brain injury and their proxies.

 I appreciate your efforts to improve the manuscript and to respond to the comments made in the first review process. However, there is still one minor issue that needs to be addressed:

Dear Reviewer, Thank you for your comments and suggestions. Below you will find our detailed response.

Detailed information:

Abstract

Point 1, Line 38-58, page 1-2: The guidelines require abstracts to be no more than 200 words, and I think this abstract is well over the limit.

Response 1: We have shortened the abstract as follows:

“The Rivermead Post-Concussion Symptoms Questionnaire (RPQ) assesses post-concussion symptoms (PCS) after traumatic brain injury (TBI). The current study examines the applicability of self-report and proxy versions of the German RPQ in adolescents (13–17 years) after TBI. We investigated reliability and validity on the total and scale score level. Construct validity was investigated by correlations with the Post-Concussion Symptoms Inventory (PCSI-SR13), Generalized Anxiety Disorder Scale 7 (GAD-7), and Patient Health Questionnaire 9 (PHQ-9) and by hypothesis testing regarding individuals’ characteristics. Intraclass correlation coefficients (ICC) assessed adolescent-proxy agreement. In total, 148 adolescents after TBI and 147 proxies completed the RPQ. Cronbach’s α (0.81– 0.91) and McDonald’s ω (0.84–0.95) indicated good internal consistency. The three-factor structure outperformed the unidimensional model. The RPQ was strongly correlated with the PCSI-SR13 (self-report: r=0.80; proxy: r=0.75) and moderately to strongly with GAD-7 and PHQ-9 (self-report: r=0.36, r=0.35; proxy: r=0.53, r=0.62). Adolescent-proxy agreement was fair (ICC[2,1]=0.44, CI95% [0.41, 0.47]). Overall, both self-report and proxy assessment forms of the German RPQ are suitable for application in adolescents after TBI. As proxy ratings tend to underestimate PCS, self-reports are preferable for evaluations. Only if a patient is unable to answer, should a proxy be used as a surrogate.”

Introduction

Point 2, Line 96-101, page 2: It might be a better choice to put these more detailed contents in the discussion section.

Response 2: Following your previous suggestion, we believe that this information is appropriate in this section, as it serves as the basis for our rationale for examining the applicability of the RPQ and its psychometric properties in adolescents. Therefore, we would prefer to keep this information in the introduction.

 Materials and Methods

Point 3, Line 201-214, page 5: I suggest moving this section to the discussion section.

Response 3: In the methods section, we provide information about the underlying analyses (e.g., examination of an alternative factor structure) to inform the reader about the form in which the RPQ was used. Therefore, we would rather keep this information in the method section.

Point 4, Line 234, 246, page 6: The abbreviations “GAD-7”, “PHQ-9” already appear in the subtitle and can be used directly.

Response 4: Adapted.

Point 5, Line 192-256, page 5-6: I recommend reducing the sentences in this section and trying to use more concise words to express more rich content.

Response 5: We have adapted the instruments descriptions as follows:

Line 171 ff., page 5: 

“Rivermead Post-Concussion Symptoms Questionnaire (RPQ)

The RPQ assesses 16 common PCS which are rated on a five-point Likert-type scale. The total score is calculated as the sum of all item ratings above the value of 1, indicating stronger impairment after TBI, and ranges from 0 (no increased difficulties since TBI) to 64 (most severe symptoms). In adult samples, the instrument has demonstrated good to excellent test-retest reliability (rtt = 0.90) [14], split-half reliability (r = 0.82 to r = 0.95), and internal consistency (Cronbach’s α: 0.89 to 0.93) [24] as well as moderate to high convergent validity and good discriminant validity [24]. The factor structure of the RPQ has been repeatedly examined [29–35] due to insufficient replication of the original one-factor structure in independent research. While there is still no consensus on the most suitable factorial structure, evidence points toward a three-factor solution including cognitive, emotional, and somatic scales [36] demonstrate a good fit both cross-sectionally (including multiple languages) [25,37] and longitudinally [38]. Therefore, in addition to the conventional RPQ total score, the analyses reported here covered the three scales.

In the present study, the self-report version as well as the newly adapted proxy-assessment form of the RPQ were administered. These two forms of the questionnaire differed solely regarding the wording in the introductory instructions (i.e., “do you suffer from...” or “does your child suffer from...”). Missing values on one to four RPQ items were imputed by prorating the scale mean, while participants with five or more missing values were excluded from further analyses.

Post-Concussion Symptom Inventory (PCSI)

The PCSI [20,21] assesses PCS by asking the individuals after TBI to rate the intensity of symptoms relative to before the injury. The age-adapted PCSI-SR13 Rapid Version [21] comprises 21 items relating to symptom severity before (pre-injury) and after (post-injury) TBI forming three scales (cognitive, emotional, and somatic). The PCSI-SR13 utilizes a seven-point Likert scale (0 = ‘not a problem at all’ to 6 = ‘serious problem’) and the total score ranges from 0 to 132 with higher values indicating more severe PCS. A retrospective-adjusted post-injury difference (RAPID) can be calculated to assess clinically significant changes between the pre- and post-injury status with a change of 80% (either improvement or worsening) considered clinically relevant. Previous research in adolescents (13–22 years) [21] has reported good to excellent internal consistency of the English version of the PCSI-SR13 total score (Cronbach’s α = 0.94) and its scales (Cronbach’s α: 0.79 to 0.93), test-retest reliability (rtt = 0.64 to rtt = 0.79), as well as evidence for convergent, predictive, and factorial validity. In the present study, the proxies filled in the respective PCSI parent-form and only results obtained from the post-injury version of the PCSI were involved in the psychometric analyses with the RPQ.

Generalized Anxiety Disorder Scale 7 (GAD-7)

The GAD-7 [21,39] is a PROM consisting of seven symptoms related to generalized anxiety disorder according to the DSM-IV [40]. Symptoms are rated on a four-point Likert scale (0 = ‘not bothered at all’ to 3 = ‘bothered nearly every day’). The total score ranges from 0 to 21 with higher values indicating greater disturbance, and scores greater than or equal to 5, 10, or 15 representing mild, moderate, or severe impairment, respectively [39]. Previous research in healthy adolescents (10–18 years) has demonstrated good to excellent internal consistency (Cronbach’s α: 0.91 to 0.95), as well as evidence of convergent, and factorial validity [41,42]. Multiple previous studies have administered the GAD-7 to assess anxiety in adolescent TBI samples, but lacked an investigation of its psychometric characteristics [43,44].

Patient Health Questionnaire 9 (PHQ-9)

The PHQ-9 [45] is as a nine-item PROM screening for major depression based on the DSM-IV [40] criteria. Items are rated on a four-point Likert scale (0 = ‘not bothered at all’ to 3 = ‘bothered nearly every day’). Total score ranges from 0 to 27 with higher values indicating more severe impairment. Scores between 1 and 4 indicate minimal depression [45,46], and values greater than or equal to 5, 10, or 15 represent mild, moderate, or severe impairment, respectively. Previous research in healthy adolescents (14–18 years) has shown good internal consistency (Cronbach’s α: 0.84) and test-retest reliability (rtt = 0.80 to rtt = 0.88), as well as convergent, criterion, and factorial validity [47,48]. Like the GAD-7, the PHQ-9 has been used in multiple studies to assess depressive symptoms in adolescents after TBI [49,50].”

Point 6, Line 271-277, page 6: Please add the reliability indicator “MacDonald’s ω”.

Response 6: We agree that McDonald’s ω is a useful indicator to assess internal consistency in addition to Cronbach’s α. Consequently, we calculated McDonald’s ω and provided the corresponding information in the methods section, Table 3, and in the results section:

Methods; Line 244, page 6:

“The reliability analyses included calculations of Cronbach's α and McDonald’s ω for the RPQ total and the scales scores, with values between 0.70 and 0.95 (Cronbach’s α) and above 0.80 (McDonald’s ω) indicating good to excellent internal consistency [52,53].”

Results; Line 335, page 9: 

“The internal consistency of the RPQ total score as well as of the three scales was good to excellent according to Cronbach’s α and McDonald’s ω values.”

Point 7, Line 309-321, page 7: This paragraph is not clear about the hypothesis, please reorganize the logic and rewrite. 

Response 7: We revised the above-mentioned paragraph as follows:

Line 282, page 7:

„We examined construct validity by testing hypotheses on concerning gender, TBI severity, and functional recovery using self-reported RPQ data. Higher RPQ values were expected in females compared to males [59], adolescents after moderate and severe TBI compared with those after mild TBI [60], and in individuals who had not fully recovered (KOSCHI < 5b) in contrast to those who showed a complete recovery (KOSCHI = 5b) [61]. In addition, we compared participants with different levels of anxiety and depression (i.e., no or minimal symptom burden: 0–4 vs. at least mild symptom burden: ≥ 5) using both self-reported and proxy assessed RPQ data. We hypothesized that higher RPQ scores would be associated with higher levels of emotional distress [60,62]. All group comparisons were performed using non-parametric Mann-Whitney-U-tests for independent samples. Cliff’s δ determined the effect size for the group comparisons with the following cut-off values: δ < |0.28| (small), 0.28 < |δ| < 0.43 (medium), and δ ≥ |0.43| (large) [63].”

 Table 4

Point 8, “χ2” and “df should be italicized, please check if the same error occurs in the full text.

Response 8: We checked and changed the font style accordingly.

Figure 2

Point 9, This image is very blurry. To give the reader a better sensory experience, please try using a higher resolution image.

Response 9: We hope that in the most recent version of the manuscript the figure will not be blurry anymore, since it was created in a high-resolution format (600 dpi, tiff). We will additionally submit the figure as a separate file to be included in the final article.

Thank you and best,

Your reviewer

Thank you!
